# Recurrence Outcome in Hepatocellular Carcinoma within Milan Criteria Undergoing Microwave Ablation with or without Transarterial Chemoembolization

**DOI:** 10.3390/medicina58081016

**Published:** 2022-07-29

**Authors:** Guobin Chen, Hong Chen, Xing Huang, Sisi Cheng, Susu Zheng, Yanfang Wu, Tanghui Zheng, Xiaochun Chen, Xinkun Guo, Zhenzhen Zhang, Xiaoying Xie, Boheng Zhang

**Affiliations:** 1Department of Hepatic Oncology, Zhongshan Hospital (Xiamen), Fudan University, Xiamen 361015, China; chen.guobin@zsxmhospital.com (G.C.); chen.hong@zsxmhospital.com (H.C.); huang.xing@zsxmhospital.com (X.H.); cheng.sisi@zsxmhospital.com (S.C.); zheng.susu@zsxmhospital.com (S.Z.); wu.yanfang@zsxmhospital.com (Y.W.); zheng.tanghui@zsxmhospital.com (T.Z.); chen.xiaochun@zsxmhospital.com (X.C.); guo.xinkun@zsxmhospital.com (X.G.); zhang.zhenzhen@zsxmhospital.com (Z.Z.); 2Xiamen Clinical Research Center for Cancer Therapy, Xiamen 361015, China; 3Department of Hepatic Oncology, Zhongshan Hospital, Fudan University, Shanghai 200032, China

**Keywords:** hepatocellular carcinoma, microwave thermotherapy, chemoembolization, recurrence

## Abstract

*Background and Objectives*: The recurrence outcome in patients who underwent microwave ablation (MWA) with or without transarterial chemoembolization (TACE) for hepatocellular carcinoma (HCC) within Milan criteria remains unclear. The aim of this retrospective study was to identify the predictive factors of recurrence in these patients. *Materials and Methods*: From May 2018 to April 2021, 66 patients with HCC within Milan criteria were enrolled. Local tumor progression (LTP) and recurrence-free survival (RFS) were evaluated. Univariate and multivariate analyses were used to evaluate the risk factors of recurrence. The propensity score analysis was conducted to reduce potential confounding bias. *Results*: During the median follow-up of 25.07 months (95% confidence interval [CI], 21.85, 28.28), the median time to LTP and RFS were 20.10 (95%CI, 14.67, 25.53) and 13.03 (95%CI, 6.36, 19.70) months. No group difference (MWA vs. MWA + TACE) was found in 1-year cumulative LTP (*p* = 0.575) and RFS (*p* = 0.515), but meaningful significant differences were found in two-year recurrence (LTP, *p* = 0.007 and RFS, *p* = 0.037). Univariate and multivariate analyses revealed that treatment received before ablation was an independent risk factor of LTP (hazard ratio [HR] 4.37, 95%CI, 1.44, 13.32) and RFS (HR 3.41, 95%CI, 1.49, 7.81). *Conclusions*: The LTP and RFS in the MWA group were similar to that in the MWA combined with TACE. For HCC within Milan criteria, both groups preferentially selected MWA. More endeavor and rigorous surveillance should be taken to relapse prevention, in patients who have received previous treatment.

## 1. Introduction

Primary liver cancer is the third leading cause of cancer death. Globalcan2020 estimated 906,000 new cases in 2020, with approximately 75–85% cases being hepatocellular carcinoma (HCC) [1]. Thermal ablation, transplantation and resection were curative therapies for patients with HCC within the Milan criterion and are the available options in the China liver cancer (CNLC) staging system [2], the European Association for the Study of the Liver (EASL) [3], and the American Association for the Study of Liver Diseases (AASLD) guidelines [4]. However, approximately 20% of patients with hepatocellular carcinoma may experience a survival benefit from resection and liver transplantation [5]. Moreover, some patients have missed out on surgery owing to poor liver function (such as severe liver cirrhosis), location of the tumor nodules or rejection of surgery [6]. Liver transplantation is also limited due to the lack of availability of liver transplants and the high cost. Therefore, there is a need for a less invasive and effective treatment method.

Microwave ablation (MWA) is a relatively new ablation technique in thermal ablation and has the demonstrated benefits of safety, as well as being effective and minimally invasive [7]. When compared to radiofrequency ablation (RFA), MWA has the advantage of being less susceptible to the heat sink effect and provides a larger ablation zone [8,9,10]. There has been no demonstrated difference between MWA and RFA in efficacy or local tumor progression [11]. A meta-analysis depicted an analogical efficacy and safety between MWA and RFA. However, MWA displayed a preponderance in reducing the rate of five-year recurrences [12]. Some retrospective studies have shown that MWA achieved long-term oncologic outcomes for ≤4 cm HCC and equivalent metastasis and recurrence rates for ≤5 cm HCC when compared with surgery [7,13]. Unfortunately, to improve the clinical outcomes, it is not enough for HCC with a single treatment.

In addition to being the first-line treatment for intermediate-stage HCC, transarterial chemoembolization (TACE) can also be used for early-stage HCC [14]. TACE combined with MWA has been shown to be an effective treatment with a mean overall survival (mOS) rate of 54.9 months in early HCC [15]. The recurrence pattern of HCC was shown to be relative to post-recurrence survival [16]. Therefore, the purpose of this study was to clarify the risk factors for recurrence in patients with HCC within the Milan criteria, receiving MWA with or without TACE.

## 2. Materials and Methods

### 2.1. Study Population

This retrospective study was approved by the ethics committees of Xiamen Branch, Zhongshan Hospital, Fudan University (authorization number B2019-010). All patients were diagnosed with HCC according to the EASL guidelines [3]. The study enrolled 66 patients with HCC who received MWA with or without TACE from May 2018 to April 2021 in our institution, as shown in Figure 1. Patients were selected according to the following criteria:

(1) HCC within the Milan criteria (single tumor ≤ 5 cm or two to three tumors ≤ 3 cm without vascular invasion) [17] who had received MWA with or without TACE as the treatment; (2) Child–Pugh score ≤ 7; (3) Eastern Cooperative Oncology Group performance status 0 or 1; and (4) refusal or unfitness for surgery and liver transplantation.

The exclusion criteria were (1) missing examination imaging from one to three-month postoperative period; (2) a lack of complete follow-up information; (3) other malignant tumors; and (4) other anti-cancer therapy received less than one month prior to intervention.

Demographics, oncology characteristics and some serological markers from within seven days before the operation were collected for analysis. The albumin–bilirubin (ALBI) score was calculated as follows: (log_10_ total bilirubin (µmol/L) × 0.66) + (serum albumin (g/L) × −0.085) [18]. The tumor location was recorded, including the tumor nodule position such as hepatic subcapsular, near large vessels, diaphragm or gallbladder. Treatment before ablation was defined as treatment up to one month prior to the intervention, such as TACE, conventional surgery, system treatment, radiotherapy or other anti-tumor treatment.

### 2.2. MWA Procedure

The tumor location and size were assessed for all patients by contrast enhanced ultrasound (CEUS) and contrast enhanced magnetic resonance imaging (CE-MRI) before the operation. Under the guidance of real-time ultrasonic imaging (US), MWA was performed with commercially available MWA systems (Covidien Medical Devices Technology Co., Ltd., Mansfield, MA, USA) after local or general anesthesia. An ablation needle with antennae was inserted into the tumor. During the MWA procedure, ablation session, time (range 1–10 min each session) and energy power (range 40–100 W each session) were determined depending on the tumor size, shape and location. The melting range was to achieve an ablated margin of at least 5 mm around the tumor. Finally, needle-path ablation was used to prevent post-operative bleeding and needle-path metastases.

### 2.3. TACE Combined with MWA Procedure

TACE and MWA procedures performed in the same session similar to Roberto Iezzi et.al commenting [19]. In brief, angiography was performed prior to ablation. All TACE procedures were performed by an experienced physician starting with a routine Seldinger puncture in the arteria cruralis after local anesthesia. A 5 French catheter (Progreat Lambda, Terumo, Japan) was selected to perform arteriography of the celiac and common hepatic artery to identify the tumor and feeders. Afterwards, MWA was performed as described above. CEUS (with sonoview as the enhancing agent) was used to evaluate the completeness of the ablation. If there was a residual tumor, an additional ablation was performed. Subsequently, a microcatheter was used to superselect the tumor-feeding branch and embolization was performed with 1–3 mL of lipiodol, with or without 10 mg of epirubicin (water-in-oil technique was used to mix chemotherapeutic agents and iodized oil which was described in previous study [20]). Finally, angiography was repeated to evaluate the extent of the lipiodol deposits. The brief procedure was shown in Appendix A.

### 2.4. Follow-Up

All patients received regular follow-up after the operation. A CE-MRI/CT was performed one to three months after MWA with or without TACE. If the tumor was completely ablated, the patient was followed-up every two to three months. Combination therapy such as immunotherapy, antiangiogenic therapy, surgery etc. would then be conducted based on tumor recurrence. The study was censored on 2 February 2022.

### 2.5. Evaluation of Therapeutic Outcomes

The treatment response was evaluated by the modified response evaluation criteria in solid tumors (mRECIST) [21]. A non-specifically trained radiologist may magnify variability in the evaluation of treatment outcomes [22]. The assessment of treatment response was calculated by two different experts, one of whom has over 10 years of experience in radiology and another who has 10 years of experience in interventional radiology. A one-month postoperative CE-MRI/CT was used for assessing the initial therapeutic effect. Local tumor progression time (LTPt) was defined as the time between the operation date and the date when any residual or new-onset tumor around the ablation zone was discovered in the same liver lobe. Recurrence-free survival (RFS) was defined as the interval from the date of MWA with or without TACE to the date of HCC recurrence. Overall survival (OS) was defined as the time from the operation time until the time of death or the last follow-up recorded. Adverse events that occurred within one week of the interventional operation were recorded at follow-up, as were complications that were considered likely to be MWA/TACE-related.

### 2.6. Statistical Analysis

All analyses were performed using SPSS 21 software (IBM Corp., Armonk, NY, USA). Data were presented as mean ± standard deviation for continuous variables, which were analyzed by an independent *t*-test similar to the previous study [23]. Categorical data were defined as frequency (as a percentage) and calculated by applying a chi-square test or Fisher’s exact test. LTP, RFS and OS were compared with a log-rank test using the Kaplan–Meier method. The risk factors for recurrence were analyzed by a univariate analysis. Multivariate Cox regression models were built to include all variables found to be *p* ≤ 0.1 in the univariate analyses. A 1:1 propensity score with logistic regression was performed for balancing variables [24], with a caliper distance of 0.1. All comparisons were two-sided, and *p* values less than 0.05 were considered statistically significant.

## 3. Results

### 3.1. Patient Characteristics

The baseline patient characteristics of the two groups are summarized in Table 1. Of the 66 patients, 87.9% patients suffered from chronic viral hepatitis B (HBV), and 75.8% suffered from hepatic cirrhosis. The majority of patients presented with BCLC Stage A (84.8%) and received other treatment before ablation (63.4%). The 44 patients who underwent MWA had a significantly smaller tumor size compared to the 22 patients who received MWA combined with TACE (18.36 ± 8.47 vs. 25.09 ± 10.31 mm, *p =* 0.006). In addition, the MWA group had a smaller number of tumor nodules (11.4% multiple nodules) according to MWA plus the TACE group (multiple nodules 11.4% vs. 36.4%, *p* = 0.038). The lymphocyte counts (1.70 ± 0.61 vs. 1.37 ± 0.43) and ALBI (−3.07 ± 0.31 vs. −3.25 ± 0.27, *p* = 0.027) of the MWA group were significantly higher than MWA plus TACE.

### 3.2. Treatment Outcomes and Complications

The rate of technical success of the ablations was 100%. During the median follow-up period of 25.07 months (95% CI, 21.85, 28.28), 6.1% of patients died. There was no significant difference between the two groups (*p* = 0.396). The estimated OS rates of the MWA group were 100% and 97% at one year and two years, respectively, and 100% and 85%, respectively, in the MWA plus TACE group (Figure 2a). A total of 37/66 (56.1%) ablations demonstrated LTP. The median LTPt (mLTPt) was 20.10 months (95% CI, 14.67, 25.53). The 1-year and 2-year cumulative LTP incidence was 40% and 53%, respectively, in the MWA group and 38% and 88%, respectively, in the MWA plus TACE group. RFS occurred in 41/66 (62.1%) patients. The median RFS (mRFS) time was 13.03 months (95% CI, 6.36, 19.70). The cumulative RFS was estimated to be 45% and 38% in the MWA group, respectively, and 61% and 15% in the MWA plus TACE group, respectively, at one year and two years. There were no significant differences found in mLTPt (*p* = 0.575, Figure 2b) and mRFS (*p* = 0.515, Figure 2c), but statistically significant differences were observed in 2-year cumulative LTP (*p* = 0.007) and RFS (*p* = 0.037) incidence.

The adverse events were evaluated by Common Terminology Criteria for Adverse Events (CTCAE) Version 5.0 (Department of Health and Humen services, USA). As shown in Table 2, 80.3% of patients experienced an adverse reaction, of which the majority were Grade ½ adverse events. Neither the Grade ½ complication rates or the Grade ¾ level adverse events were significantly different between the MWA group and the MWA plus TACE group, ½ (*p* = 0.595) and ¾ (*p* = 0.735), respectively.

### 3.3. Univariable and Multivariable Analyses for LTP and RFS

In the univariate analysis, Barcelona Clinic Liver Cancer (BCLC) stage (*p* = 0.097), tumor location (*p* = 0.086), treatment before ablation (*p* < 0.001), baseline lymphocyte counts (*p* = 0.024), baseline monocyte count (*p* = 0.034), TNF-α (*p* = 0.050) and ALBI (*p* = 0.064) were taken into multivariate Cox models for LTP. As shown in Table 3, receiving treatment before ablation (HR 4.37, 95% CI, 1.44, 13.32) and baseline ALBI (HR 4.31, 95% CI, 1.17, 15.92) independently predicted the LTPt. As shown in Table 4, treatment before ablation was associated with RFS (*p* = 0.003) and was verified as an independent predictor for RFS by multivariate analysis (HR 3.41, 95% CI, 1.49, 7.81).

### 3.4. Subgroup Analysis

In the subgroup analysis, 29/66 (43.9%) and 24/66 (36.4%) of patients had been treated with TACE or traditional surgery, respectively, at least one month before ablation. The univariate analysis showed that TACE (*p* < 0.003 and *p* = 0.001, respectively) and radiotherapy (*p* < 0.001) were associated with LTP and RFS. In the multivariate analysis, TACE (HR 3.92, 95% CI, 1.72, 8.93) and radiotherapy (HR 17.95 95% CI, 4.10, 78.71) were independent predictors of LTP. The risk factors for RFS were antiangiogenic therapy (HR 2.54, 95% CI, 0.99, 8.93) and radiotherapy (HR 8.41, 95% CI, 2.29, 30.89) (Table 5).

The rate of LTP and RFS was compared between the MWA and MWA plus TACE group after propensity score matching. As depicted in Appendix A, for mLTPt (*p* = 0.945) and mRFS (*p* = 0.28), there were still no significant differences between the two groups after balancing the variables.

## 4. Discussion

In the present study, we demonstrated that recurrence and major complication rates of the MWA group were similar to the MWA combined with TACE group, meeting the Milan criteria in terms of LTP and RFS. Furthermore, treatment before ablation in particular with TACE, antiangiogenic therapy and radiotherapy were independent risk factors for tumor recurrence.

MWA is a promising thermal technique because of its efficacy and safety. A meta-analysis proved that MWA had a relative risk of 0.93 (95% CI, 0.78, 1.14) compared to RFA for 1-year LTP in early HCC [25]. In previous research, 729 patients with HCC within the Milan criteria undergoing MWA or surgical resection (SR) were analyzed retrospectively. They identified that MWA achieved comparable long-term oncologic outcomes such as LTP or disease-free survival (DFS) with SR for ≤4 cm HCC [7]. A randomized controlled trial that screened 278 patients with 3–5 cm HCC reported that the one-year recurrence rates in the TACE combined with MWA group was significantly lower than the MWA or TACE groups [26]. However, TACE was followed by MWA after 15 days in this study. TACE combined with simultaneous DynaCT-guided MWA was reported as an outstanding method for the treatment of <5 cm HCC in contrast to TACE [27]. With this method, the mean PFS was 28.22 months longer than the TACE group. Furthermore, a single tumor less than 3 cm showed a prolonged PFS and OS when performed by the TACE combined with MWA [15]. To our knowledge, the recurrence outcome of MWA with or without simultaneous TACE has yet to be researched in depth. Moreover, few studies have focused on all HCCs meeting the Milan criteria.

In the current study, the mLTPt was greater than in another recent study (20.10 vs. 9.60 months) [28], which is of note particularly because we did not exclude patients who relapsed after receiving other antitumor therapies before ablation. Univariate and multivariate analyses revealed that treatment before ablation was a significant risk factor for LTP and RFS. As shown in the subgroup analysis, patients with TACE, antiangiogenic therapy or radiotherapy demonstrated earlier relapse. As described in a previous study, TACE was an independent risk factor for worse PFS when used to treat HCC 5 cm or smaller [29,30]. Moreover, Salas et al. found that radiotherapy was a key factor influencing the incidence of local recurrence in solitary fibrous tumors [31]. To the best of our knowledge, the present study is the first to explore occurrence outcomes of HCC treated with MWA with or without TACE. Patients who received treatment before ablation tended to possess poor tumor characteristics, and treatment such as radiotherapy may increase VEGF/plt level, which is associated with poor outcomes [32]. This may explain the recurrence of LTP and RFS.

The treatment of MWA combined with TACE seemed to have a similar result to MWA with regard to LTP and RFS in our results. For LTP and RFS at two years, the late relapse rate of the combined therapy group was significantly worse than the MWA group. However, there were differences in the baseline variables relating to tumor size, nodule number, lymphocyte counts and ALBI. Tumor size is one of the critical factors for complete ablation and survival. For a single tumor nodule ≤3 cm, priority should be given to MWA over treatment with TACE [29]. However, for lesions in the range 3.1–5 cm, this study demonstrated that MWA had similar effects as TACE in OS. A multicenter observational study demonstrated that tumor size and number were crucial prognostic factors for HCC with TACE [33]. Another study has also shown that lymphocyte counts and ALBI play an important role in the carcinogenesis and progression of HCC. In our research, the combination therapy group had larger and more numerous tumor nodules; moreover, lymphocyte counts and ALBI were higher than in the MWA group [34,35]. This appears to suggest that the combination treatment was superior for the prevention of tumor recurrence. However, after balancing the variables, this was not validated. This may have been due to the small sample size. Further studies are required to explore the possibility. Howsoever, MWA will be effective in HCC within Milan criteria and the single procedure doesn’t meet an early relapse.

No serious adverse events (Grade 4) were recorded for the entire follow-up period. However, 18.2% patients suffered Grade 3 events; all of them recovered with symptomatic treatment. This indicated that both treatment methods were safe for patients.

Some limitations were identified in the study. First, it was a single center retrospective study with a small sample, and the two groups differed significantly on some variables. Selection bias is inescapable; however, we used the propensity matching score to lessen the effect. Second, 87.9% of our study cohort had an HBV infection; different etiologies may influence tumor characteristics. Thirdly, ultrasound was used to ensure complete ablation in the study. However, post-ablation ultrasound images may be affected by gas or inflammatory edema around the ablation site. Three dimensional digital subtraction and angiography technology may eliminate the influence as previous study have reported [15,27]. This deserves further exploration. Finally, the difference of intraoperative medications and procedures in MWA and TACE may have affected the outcome of treatment. To minimize the differences, all procedures were performed by the same team. Therefore, further research with a large stratified multicenter patient cohort is necessary to validate our results.

## 5. Conclusions

In short, the LTP and RFS in the MWA group were comparable to that in the group treated with MWA combined with TACE. In our results, for HCC meeting the Milan criteria, priority should be given to MWA when making treatment choices between MWA and MWA combined with TACE. The approach possesses further relevant advantages such as the decrease of patient discomfort and cost savings due to unnecessary TACE is not required. Moreover, receiving treatment before ablation was an independent risk factor for recurrence. When patients receive prior treatment, more rigorous surveillance should be taken to closely observe them for recurrence. Besides, further prospective studies with larger samples are required to clarify the distinction between various treatments for HCC.

## Figures and Tables

**Figure 1 medicina-58-01016-f001:**
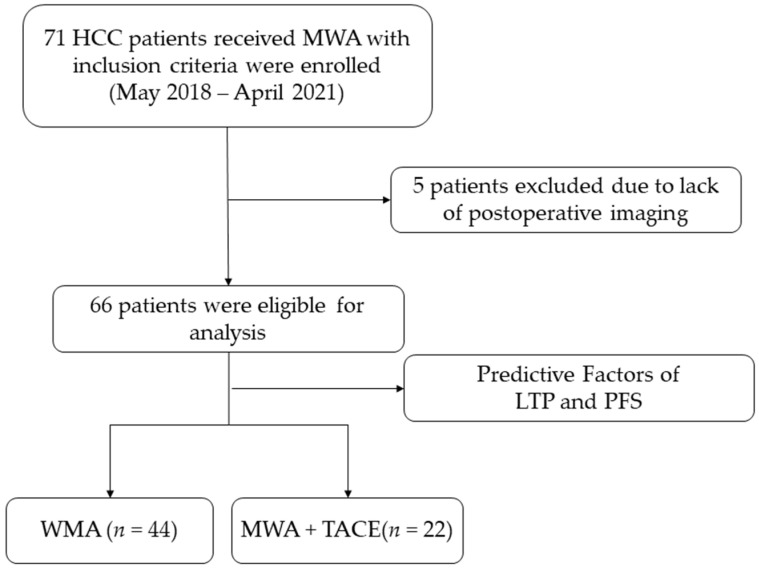
Study flowchart.

**Figure 2 medicina-58-01016-f002:**
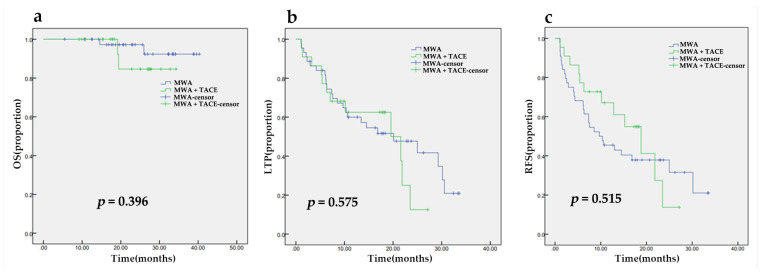
The Kaplan–Meier analysis of overall survival (**a**), time to local tumor recurrence (**b**) and recurrence-free survival (**c**) after MWA with or without TACE.

**Table 1 medicina-58-01016-t001:** Baseline patient characteristics stratified by therapy [mean SD/N (%)].

Characteristics	Overall (N = 66)	MWA (N = 44)	MWA + TACE (N = 22)	*p* Value
age	61.14 ± 11.18	61.36 ± 10.89	60.68 ± 11.98	0.817
Sex				
male	51 (77.3%)	31 (70.5%)	20 (90.9%)	0.062
female	15 (22.7%)	13 (29.5%)	2 (9.1%)	
Hepatic Cirrhosis				
yes	50 (75.8%)	34 (77.3%)	16 (72.7%)	0.685
no	16 (24.2%)	10 (22.7%)	6 (27.3%)	
Diabetes				
yes	12 (18.2%)	8 (18.2%)	4 (18.2%)	>0.999
no	54 (81.8%)	36 (81.8%)	18 (81.8%)	
Hypertension				
yes	17 (25.8%)	13 (29.5%)	4 (18.2%)	0.320
no	49 (74.2%)	31 (70.5%)	18 (81.8%)	
HBV				
yes	58 (87.9%)	39 (88.6%)	19 (86.4%)	>0.999
no	8 (12.1%)	5 (11.4%)	3 (13.6%)	
Tumor diameter (mm)	20.61 ± 9.59	18.36 ± 8.47	25.09 ± 10.31	0.006
BCLC				
A	56 (84.8%)	39 (88.6%)	17 (77.3%)	0.396
B	10 (15.2%)	5 (11.4%)	5 (22.7%)	
Tumor location				
special location	48 (72.7%)	33 (75%)	15 (68.2%)	0.716
traditional location	18 (27.3%)	11 (25%)	7 (31.8%)	
Tumor number				
single	53 (80.3%)	39 (88.6%)	14 (63.6%)	0.038
multiple	13 (19.7%)	5 (11.4%)	8 (36.4%)	
Treatment before ablation				
yes	24 (36.4%)	18 (40.9%)	6(27.3%)	0.278
no	42 (63.6%)	26 (59.1%)	16(72.7%)	
Baseline AFP(ng/mL)	433.81 ± 1666.24	285.50 ± 1219.54	730.44 ± 2325.90	0.310
Baseline Lymphocytes (×109/L)	1.59 ± 0.58	1.70 ± 0.61	1.37 ± 0.43	0.034
Baseline Monocytes(×109/L)	0.50 ± 0.17	0.51 ± 0.17	0.47 ± 0.18	0.452
Baseline ALB (g/L)	44.51 ± 3.21	43.86 ± 3.18	45.77 ± 2.94	0.022
Baseline ALT (U/L)	28.49 ± 14.91	30.61 ± 16.70	24.36 ± 9.64	0.061
Baseline AST (U/L)	30.06 ± 13.79	31.56 ± 15.90	27.14 ± 7.77	0.137
Baseline rGT (U/L)	60.56 ± 53.02	64.48 ± 56.12	52.73 ± 46.44	0.400
Baseline LDH(U/L)	197.89 ± 44.45	192.88 ± 43.41	207.68 ± 45.83	0.207
CD4/CD8	1.98 ± 1.21	2.04 ± 1.38	1.87 ± 0.85	0.626
IL-6 (pg/mL)	6.61 ± 7.11	7.11 ± 8.57	5.65 ± 2.71	0.474
TNF-α (pg/mL)	12.07 ± 12.21	10.43 ± 11.14	15.09 ± 13.77	0.203
Baseline ALBI	−3.13 ± 0.31	−3.07 ± 0.31	−3.25 ± 0.27	0.027
Baseline CRP(mg/L)	3.10 ± 8.28	6.85 ± 16.84	2.13 ± 2.38	0.627

Median with standard deviation are shown for quantitative variables and counts with proportions are shown for categorical variables. Tumor special location including tumor nodule in the position such as hepatic subcapsular, near large vessels, diaphragm and gallbladder. Abbreviations: Ref-Reference; HBV-Hepatitis B Virus infection; AFP-alpha-fetoprotein; ALB-albumin; ALT-alanine transaminase; AST-aspartate aminotransferase; γGT-γ-glutamyltranspeptidas; LDH-lactate dehydrogenase; ALBI-albumin–bilirubin; CRP-c-reactive protein.

**Table 2 medicina-58-01016-t002:** Adverse Events and Complications.

Categories	Overall (N = 66)			MWA (N = 44)		MWA + TACE (N = 22)	
Grade	1–4 level	½ level	¾ level	½ level	¾ level	½ level	¾ level
Adverse events	53 (80.3%)	52 (78.8%)	12 (18.2%)	36 (81.8%)	7 (15.9%)	16 (72.7%)	5 (22.7%)
Fever	2 (3.0%)	2 (3.0%)	0 (0%)	1 (2.3%)	0 (0%)	1 (4.5%)	0 (0%)
Nausea or vomiting	13 (19.7%)	13 (19.7%)	0 (0%)	8 (18.2%)	0 (0%)	5 (22.7%)	0 (0%)
Fatigue	5 (7.6%)	5 (7.6%)	0 (0%)	2 (4.5%)	0 (0%)	3 (13.6%)	0 (0%)
Abdominal pain/distension	4 (6.1%)	4 (6.1%)	0 (0%)	3 (6.8%)	0 (0%)	1 (2.3%)	0 (0%)
Total bilirubin elevation, transient	1 (1.5%)	1 (1.5%)	0 (0%)	1 (2.3%)	0 (0%)	0 (0%)	0 (0%)
ALT elevation	45 (68.2%)	42 (63.6%)	3 (4.5%)	28 (63.6%)	2 (4.5%)	14 (63.6%)	1 (2.3%)
AST elevation	53 (80.3%)	43 (65.2%)	10 (15.2%)	30 (68.2%)	7 (15.9%)	13 (59.1%)	3 (13.6%)

**Table 3 medicina-58-01016-t003:** Univariable and multivariable predictors of LTP.

Characteristics	Univariate Analysis		Multivariate Analysis	
	HR (95% CI)	*p* Value	HR (95% CI)	*p* Value
age	0.99 (0.96, 1.02)	0.571		
Sex				
male	1.00 (Ref)	0.146		
female	0.50 (0.19, 1.28)			
Hepatic Cirrhosis				
yes	1.00 (Ref)	0.166		
no	1.68 (0.81, 3.52)			
Diabetes				
yes	1.00 (Ref)	0.455		
no	0.75 (0.35, 1.60)			
Hypertension				
yes	1.00 (Ref)	0.638		
no	1.20 (0.56, 2.54)			
HBV				
yes	1.00 (Ref)	0.845		
no	0.92 (0.38, 2.22)			
Tumor diameter (mm)	1.01 (0.97, 1.04)	0.768		
BCLC				
A	1.00 (Ref)	0.097	1.00 (Ref)	0.845
B	1.96 (0.89, 4.33)		1.11 (0.40, 3.04)	
Tumor location				
special location	1.00 (Ref)	0.086	1.00 (Ref)	0.181
traditional location	0.53 (0.25, 1.10)		0.55 (0.23, 1.32)	
Tumor number				
Single	1.00 (Ref)			
Multiple	1.46 (0.68, 3.14)	0.332		
Treatment before ablation				
no	1.00 (Ref)	<0.001	1.00 (Ref)	0.009
yes	4.77 (2.05, 11.07)		4.37 (1.44, 13.32)	
Baseline AFP (ng/mL)	1.00 (1.00, 1.00)	0.564		
Baseline Lymphocytes (×109/L)	0.47 (0.24, 0.90)	0.024	1.32 (0.52, 3.39)	0.561
Baseline Monocytes (×109/L)	0.10 (0.01, 0.84)	0.034	0.07 (0.00, 1.45)	0.086
Baseline ALB (g/L)	0.93 (0.85, 1.03)	0.167		
Baseline ALT (U/L)	0.99 (0.97, 1.02)	0.489		
Baseline AST (U/L)	1.00 (0.97, 1.02)	0.732		
Baseline rGT (U/L)	1.00 (0.99, 1.01)	0.798		
Baseline LDH (U/L)	1.01 (0.99, 1.01)	0.19		
CD4/CD8	0.89 (0.66, 1.20)	0.444		
IL-6 (pg/mL)	0.98 (0.92, 1.04)	0.448		
TNF-α (pg/mL)	1.03 (1.00, 1.06)	0.05	1.03 (0.99, 1.05)	0.071
Baseline ALBI	2.57 (0.95, 6.95)	0.064	4.31 (1.17, 15.92)	0.028
Baseline CRP (mg/L)	0.98 (0.94, 1.03)	0.984		

Median with standard deviation are shown for quantitative variables and counts with proportions shown for categorical variables. Tumor special location including tumor nodule in the position such as hepatic subcapsular and near large vessels, diaphragm or gallbladder. Abbreviations: Ref-Reference; HBV-Hepatitis B Virus infection; AFP-alpha-fetoprotein; ALB-albumin; ALT-alanine transaminase; AST-aspartate aminotransferase; γGT-γ-glutamyltranspeptidase; LDH-lactate dehydrogenase; ALBI-albumin–bilirubin; CRP-c-reactive protein.

**Table 4 medicina-58-01016-t004:** Univariable and multivariable predictors of RFS.

Characteristics	Univariate Analysis		Multivariate Analysis	
	HR (95% CI)	*p* Value	HR (95% CI)	*p* Value
age	0.98 (0.95, 1.01)	0.139		
Sex				
male	1.00 (Ref)	0.121		
female	1.99 (0.83, 4.76)			
Hepatic Cirrhosis				
yes	1.00 (Ref)	0.755		
no	1.12 (0.55, 2.30)			
Diabetes				
yes	1.00 (Ref)	0.455		
no	0.75 (0.36, 1.59)			
Hypertension				
yes	1.00 (Ref)	0.162		
no	1.70 (0.81, 3.59)			
HBV				
yes	1.00 (Ref)	0.734		
no	1.16 (0.49, 2.78)			
Tumor diameter (mm)	0.98 (0.95, 1.02)	0.38		
BCLC				
A	1.00 (Ref)	0.373		
B	1.42 (0.66, 3.09)			
Tumor location				
special location	1.00 (Ref)	0.705		
traditional location	0.87 (0.44, 1.76)			
Tumor number				
single	1.00 (Ref)	0.802		
multiple	1.10 (0.52, 2.31)			
Treatment before ablation				
no	1.00 (Ref)	0.003	1.00 (Ref)	0.004
yes	3.10 (1.47, 6.54)		3.41 (1.49, 7.81)	
Baseline AFP (ng/mL)	1.00 (1.00, 1.00)	0.425		
Baseline Lymphocytes (×109/L)	0.57 (0.31, 1.05)	0.07	0.95 (0.43, 2.11)	0.896
Baseline Monocytes (×109/L)	0.16 (0.02, 1.13)	0.067	0.21 (0.02, 2.51)	0.219
Baseline ALB (g/L)	0.97 (0.88, 1.06)	0.447		
Baseline ALT (U/L)	1.00 (0.98, 1.03)	0.809		
Baseline AST (U/L)	1.00 (0.98, 1.02)	0.904		
Baseline rGT (U/L)	1.00 (0.99, 1.01)	0.430		
Baseline LDH (U/L)	1.00 (0.99, 1.01)	0.984		
CD4/CD8	0.86 (0.59, 1.26)	0.447		
IL-6 (pg/mL)	0.98 (0.92, 1.04)	0.491		
TNF-α (pg/mL)	1.02 (0.99, 1.04)	0.295		
Baseline ALBI	1.86 (0.73, 4.79)	0.196		
Baseline CRP (mg/L)	0.98 (0.93, 1.03)	0.435		

Median with standard deviation are shown for quantitative variables and counts with proportions shown for categorical variables. Tumor special location including tumor nodule in the position such as hepatic subcapsular, near large vessels, diaphragm and gallbladder. Abbreviations: Ref-Reference; HBV-Hepatitis B Virus infection; AFP-alpha-fetoprotein; ALB-albumin; ALT-alanine transaminase; AST-aspartate aminotransferase; γGT-γ-glutamyltranspeptidase; LDH-lactate dehydrogenase; ALBI-albumin–bilirubin; CRP-c-reactive protein.

**Table 5 medicina-58-01016-t005:** Subgroup analysis of the treatment before ablation [N/(%)].

Method		LTP		PFS	
	Number	HR (95% CI)	*p* Value	HR (95% CI)	*p* Value
**Univariate Analysis**					
TACE	29 (43.9%)	4.56 (2.14, 9.75)	<0.001	2.27 (1.18, 4.35)	0.014
Thermal ablation	17 (25.8%)	1.43 (0.98, 2.08	0.063	1.73 (0.86, 3.48)	0.124
Antiangiogenic therapy	7 (10.6%)	2.19 (1.38, 3.46)	0.001	3.54 (1.45, 8.62)	0.005
Surgery	24 (36.4%)	1.05 (0.75, 1.47)	0.775	1.10 (0.58, 2.09)	0.765
Radiotherapy	3 (4.5%)	4.08 (1.98, 8.41)	<0.001	8.33 (2.27, 30.54)	0.001
**Multivariate Analysis**					
TACE	29 (43.9%)	3.92 (1.72, 8.93)	0.001	1.96 (1.00, 3.88)	0.053
Thermal ablation	17 (25.8%)	1.05 (0.43, 2.58)	0.912	——	——
Antiangiogenic therapy	7 (10.6%)	2.59 (0.93, 7.23)	0.068	2.54 (0.99, 6.44)	0.049
Radiotherapy	3 (4.5%)	17.95 (4.10, 78.71)	<0.001	8.41 (2.29, 30.89)	0.001

Only the variables found to be *p* ≤ 0.1 in the univariate analyses were taken into Multivariate Cox regression models.

## Data Availability

The data presented in this study are available from the corresponding authors. The data are not publicly available due to privacy restrictions.

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
