# Peer review of "Recurrence Outcome in Hepatocellular Carcinoma within Milan Criteria Undergoing Microwave Ablation with or without Transarterial Chemoembolization"

_medicina, 2022, doi:10.3390/medicina58081016_

Round 1
Reviewer 1 Report
TITLE
The title is clear and direct in its present form.
ABSTRACT
The abstract is not well structured in all its section, and therefore it does not properly reflect the main text, not highlighting the most important aspects of this paper. Therefore, some adjustments are needed. For example: in the background and objective section there is no background. In the Methods it is important to report the study design (prospective, retrospective…..).
KEYWORDS
Authors did not correctly report all keywords from MeSH Browser. In particular, for example, I checked “microwave ablation” on MeSH Browser and I did not find this KW. This is important, in my personal opinion, in order to increase the traceability of this paper (and consequently the possibility of the Journal to be cited by Readers and Stakeholders). I suggest the check of all KWs.
INTRODUCTION
Although the introduction fits the context of the study, it could be improved. In fact, many concepts clearly explicated in an exhaustive introduction could help readers to become passionate about reading the paper and using it as a reference.
In my opinion, the lines 38-49 are too long in talking about LT and resections. They could be reduced.
In the subsequent lines it is not clear the reasons why to use MW respect to RF. Did you prefer to utilize MW according to “Cancers (Basel). 2020;12(12):3796. Published 2020 Dec 16. doi:10.3390/cancers12123796”. If yes, the authors could explain this point in the text.
In the subsequent lines it is not clear the need of combination of MW plus TACE.
Furthermore, at the end of in the introduction there is a repetition of an entire sentence.
But, most of all, in the introduction there is no aim of the study.
MATERIALS & METHODS
Lines 67-68: how did you performed the diagnosis? According to the EASL? Or according to AASLD? In the AASLD GL LI-RADS were introduced. This system presented many advantages and many problems [Hepatology. 2022;10.1002/hep.32494. doi:10.1002/hep.32494 ---- Eur J Gastroenterol Hepatol. 2019;31(3):283-288. doi:10.1097/MEG.0000000000001269]. Please, could the authors explain, clearly, their modus operandi?
In the section 2.1 it is not clear what was the protocol you used. In other words: the patients were all into the Milan criteria: how did you choose to perform only MW or MW plus TACE?
Section 2.3
It is not clear how did you performed the combined treatment: could you detail it according to “Radiology. 2014;272(2):612-613. doi:10.1148/radiol.14140678”, if you agree.
Please, it is not clear the technique of TACE performed. Please, could you detail it according to “Ann Hepatol. 2021;22:100278. doi:10.1016/j.aohep.2020.10.006”, if you agree?
Section 2.4 – 2.5
It is well known that the decision on whether an HCC patient is a responder or progressor after a treatment may vary among different radiologists, especially in case of a non-specifically trained radiologist and, therefore, regardless of the adopted criteria, patients should be evaluated by experienced radiologists to minimize variability in this critical instance [Eur Radiol. 2018;28(9):3611-3620. doi:10.1007/s00330-018-5393-3]. Please, could the Authors report the methodology used: one or more experienced radiologists evaluated imaging prior to a re-treatment? one or more experienced radiologists evaluated imaging according to mRECIST? Please, cite the aforementioned paper [Eur Radiol. 2018;28(9):3611-3620. doi:10.1007/s00330-018-5393-3].
RESULTS
Results section is very clear and well structured. therefore, no further adjustments are needed.
DISCUSSION
In my opinion, the discussion could have a more clinical slant. the authors could report the clinical implications of their work. Could the authors discuss this theme: these results are related to lesions with dimensions of about overall 21 mm. In Europe and USA, the surveillance program and also the future strategy to improve the surveillance program and diagnosis of HCC will allow to overcome the ultrasound limitations. The first consequence will be the detection of more and more lesions in very early and early stage (small lesions). Could the Authors discuss these themes, and report if their proposal of MW will be effective also for small lesions in this future scenario?
Please, the authors must implement the limitation section.
TABLES & FIGURES
Tables & Figures are satisfactory, and they correctly match the quality standard of this Journal.
Personally, I prefer if the authors include a figure with MW compared with a same case with combined treatment.
Reviewer 2 Report
The manuscript is of great interest and reflects interesting and careful research. However, some aspects should be taken into account before possible publication.
1) Introduction: it is too short. The authors should increase the information and its depth.
2) Material & Methods: an outline of the procedure followed would be interesting and would give a better understanding of the article.
Line 107: there is a double space between two words. Revise.
Line 133: it is necessary to indicate the type of software (brand, country of origin). There are no references to the statistical study used? Why have these tests been used? Has the normality of the data not been studied?
Figure 2 looks too small.
3) Conclusions: they are very brief. What are the future perspectives of this study? Impact?
4) References are scarce. Look for more studies that may be of interest.
Author Response
Please see the attachment, thak you.

Round 2
Reviewer 1 Report
the authors must improve the response number 8 as suggested to the Editor.
the authors stated that they used EASL criteria and AALSD criteria to achive HCC diagnosis. however, AALSD criteria included LI-RADS. LI-RADS has many limitations (Eur J Gastroenterol Hepatol. 2019;31(3):283-288. doi:10.1097/MEG.0000000000001269). furthermore, recently it was demonstrated that agreement for LI-RADS categorization was lower for extracellular and hepatospecific contrast agents, and including LI-RADS ancillary features did not improve agreement (Hepatology. 2022;10.1002/hep.32494. doi:10.1002/hep.32494). therefore, it is very important that the authors improve this section.
for example, the authors can write that they did not use LI-RADS due to the limitations of the system [Eur J Gastroenterol Hepatol. 2019;31(3):283-288. doi:10.1097/MEG.0000000000001269 -- Hepatology. 2022;10.1002/hep.32494. doi:10.1002/hep.32494]
Author Response
Please see the attachment.

This manuscript is a resubmission of an earlier submission. The following is a list of the peer review reports and author responses from that submission.